# The Relationship between Chronotype, Physical Activity and the Estimated Risk of Dementia in Community-Dwelling Older Adults

**DOI:** 10.3390/ijerph17103701

**Published:** 2020-05-24

**Authors:** Ngeemasara Thapa, Boram Kim, Ja-Gyeong Yang, Hye-Jin Park, Minwoo Jang, Ha-Eun Son, Gwon-Min Kim, Hyuntae Park

**Affiliations:** 1Department of Healthcare and Science, Dong-A University, Busan 49315, Korea; ngeemasara@gmail.com (N.T.); sky940702@naver.com (J.-G.Y.); mihd3987@hanmail.net (H.-J.P.); mtow0620@gmail.com (M.J.); haundl123@naver.com (H.-E.S.); rlarnjsals47@gmail.com (G.-M.K.); 2Department of Neurology, Dong-A University Hospital, Busan 49201, Korea; lallal91@naver.com

**Keywords:** chronotype, dementia, ageing, depression, physical activity, tri-axial accelerometer

## Abstract

Our study examined the association between chronotype, daily physical activity, and the estimated risk of dementia in 170 community-dwelling older adults. Chronotype was assessed with the Horne–Östberg Morningness–Eveningness Questionnaire (MEQ). Daily physical activity (of over 3 METs) was measured with a tri-axial accelerometer. The Korean version of the Mini-Mental State Examination (K-MMSE) was used to measure the estimated risk of dementia. The evening chronotype, low daily physical activity, and dementia were positively associated with each other. The participants with low physical activity alongside evening preference had 3.05 to 3.67 times higher estimated risk of developing dementia, and participants with low physical activity and morning preference had 1.95 to 2.26 times higher estimated risk than those with high physical activity and morning preference. Our study design does not infer causation. Nevertheless, our findings suggest that chronotype and daily physical activity are predictors of the risk of having dementia in older adults aged 70 years and above.

## 1. Introduction

Dementia is a syndrome that affects memory, thinking, behavior, and the ability to perform daily activities, and the risk of developing dementia doubles every five years after the age of 65 [1]. It leads to a gradual decline in independent function, affecting not only the individual but also their family and healthcare system [2]. According to the World Health Organization (WHO), 50 million people worldwide have dementia, and this number is estimated to reach 152 million in 2050 [3]. Dementia results from a variety of diseases that primarily or secondarily affect the brain, such as Alzheimer’s disease (AD) stroke or other neurodegenerative diseases, and Alzheimer’s is the most common type of dementia among older adults [3,4].

One of the common behavior alterations in individuals with AD is an abnormal rest–activity pattern in daily life [5]. Activity measured using wrist-worn accelerometers has demonstrated a phase delay in the diurnal pattern of the level of activity as well as decreased activity levels during daytime in individuals with AD compared with healthy individuals [6,7,8]. The phase delay in the activity may be related to the exacerbation of the behavioral symptoms of AD in the afternoon and evening [9]. In addition, one study has shown a phase delay in Bmal1 mRNA, which is a circadian clock gene mRNA [10]. However, to the best of our knowledge, it is not clear whether phase delay in the circadian clock could contribute to the pathogenesis of AD.

To epidemiologically investigate the phases of the central circadian clock (the central clock that drives circadian rhythms of behavior and physiology) in the suprachiasmatic nucleus (SCN) during daily life, individual diurnal preferences for morning or evening activity (i.e., chronotype or morningness–eveningness, the degree to which people prefer to be active in the morning or the evening) were assessed using the Horne–Östberg Morningness–Eveningness Questionnaire (MEQ) [11]. Chronotype, or morningness–eveningness, appears to be associated with the circadian phase. For example, the biological circadian rhythm cortisol levels [12] and body temperature [13,14] along with the sleep–wake cycle on free days [15] are delayed in individuals with evening preference (evening types) compared with individuals with morning preference (morning types). Because the preference is partly associated with genetic factors [16], the score also represents the circadian trait of individuals.

Sleep and circadian rhythm disturbances are common in neurodegenerative diseases such as dementia [17]. Circadian timing mainly affects memory and cognitive function [18]. Individuals have different chronotypes, and cognitive performances differ according to an individual’s preferred chronotype (evening/morning) [19]. However in older adults, a study reported that different chronotypes show different levels of cognitive function, and the likelihood of cognitive impairment for the morning chronotype is low (after adjusting for education level) [20]. The studies are still not adequate to provide strong evidence regarding the relationship between chronotype and dementia.

A systematic review and meta-analysis of longitudinal studies shows that higher levels of physical activity reduce cognitive decline as well as the risk of dementia [21]. Regular physical activity in elderly people may also be an important factor to prevent cognitive decline and dementia [22]. In a previous study, habitual physical activity estimated using a self-administered questionnaire was positively associated with a greater tendency to be a morning type in young adults [23], indicating that a clearer understanding is also needed about the associations of the diurnal preference and physical activity with cognitive decline and/or dementia. Thus, in this study, we aimed to investigate the independent associations of the diurnal preference (chronotype) and daily physical activity with the estimated risk of dementia in nondemented community-dwelling older adults. We also investigated the combined effect of chronotype and physical activity on dementia.

## 2. Materials and Methods

### 2.1. Subjects

In this cross-sectional study, we screened 412 community-dwelling older adults who were ethnically Korean and aged ≥70 years. Of the 412 participants, 170 were selected for the study and 242 were deemed ineligible based on the exclusion criteria (Figure 1). Among the final participants, 102 were female, and 68 were male. All participants signed informed written consent and the study procedure was approved by the Institutional Review Board (IRB).

### 2.2. Chronotype and Estimated Risk of Dementia

The chronotype data was assessed using the MEQ scale [11]. The MEQ consisted of a 19-item self-report questionnaire based on individuals’ activity preferences. The participants were assigned to morning- or evening-type groups based on their MEQ scores (16 to 68). The lower scores (≤41) indicated an evening type whereas the higher scores (≥59) indicated a morning type. The participants underwent a short Korean version of the Mini-Mental State Examination (K-MMSE) [24] for the assessment of their estimated risk of dementia symptoms. The K-MMSE has a total score of 30 points, and the test consists of time orientation (5 points), spatial orientation (5 points), memory recall (3 points), language (16 points), and space-time configuration (1 point). Lower scores indicate a higher estimated risk of dementia. Depressive symptoms were also assessed using the Geriatric Depression Scale (GDS) [25]. The GDS is a 30-item self-reported questionnaire used to assess depressive symptoms in the elderly, but for this study we used a short version of the GDS test (15-item).

### 2.3. Physical Activity and Function

Daily physical activity was objectively measured through moderate-to-vigorous physical activity (MVPA) of intensity over 3 METs with a tri-axial accelerometer (Fitmit INC, Suwon, Korea). The accelerometer was worn on the non-dominant hand by the participants. A short performance physical battery (SPPB) [26] consisting of a series of physical performance tests was used to assess gait speed (4-m), balance, and strength, and endurance in the lower extremity (chair rise task). The total SPPB score ranged from 0 to 12 points (0–4 points for each test). A higher score indicated higher physical function.

(i)Gait speed test: This test included a 1.5-m acceleration distance, 4-m at “preferred walking speed”, followed by a 1.5 m deceleration distance. Only the 4-m walk was timed. Since the participants were older adults, they were accompanied by a researcher throughout the walk for their safety.(ii)Balance test: The participant’s ability to maintain balance was evaluated with three different standing positions, for 10 s each: side-by-side stance, semi-tandem stance, and a full-tandem stance. A rectangular mat traced with footprints to guide the balance position was used for the test.(iii)Chair rise task: The participant’s speed of standing up from a chair and sitting back down consecutively five times was timed. Participants were instructed to cross their arms in front of their chest during the test procedure.

Muscle strength was analyzed with a grip test using a digital hand dynamometer (TKK 5101 Grip-D, Takei, Tokyo, Japan). During the test, participants were instructed to maintain their shoulders slightly apart from their body and hold the dynamometer pointing to the ground. The test was repeated two times on both the right and left hand alternatively. All participants were encouraged to perform their best during the grip test for the best result. The EuroQol-5 Dimension (EQ-D5), which measures the health-related quality of life, was measured as a covariate.

We defined frailty as per Linda Fried’s frailty phenotype guidelines [27]. The frailty phenotype consisted of five criteria:a.Weakness—grip strength adjusted for gender and BMI;b.Weight loss—unintentional weight loss of 4.5 kg in prior year;c.Slowness—gait speed was calculated by time taken to walk without help for 4-m distance twice;d.Exhaustion—self-report of exhaustion;e.Low dose of physical activity—based on the data obtained from the accelerometer, the ACSM’s elderly physical activity recommendation.

### 2.4. Statistical Analysis

All statistical analyses were performed using IBM SPSS V21.0 software (IBM Corp., Armonk, NY, USA). The data were interpreted as mean ± standard deviation. The significance level was set to be <0.05. Student’s *t*-test analyzed the difference in baseline variables (demographic, anthropometric, physical functions, MMSE scores, and chronotype) between men and women. The physical functions and estimated risk of dementia were summarized according to chronotype (i.e., MEQ score) and differences in means between the two groups were analyzed by *t*-test for normal distribution as well as Mann–Whitney U test for non-normal distribution. Partial correlation analysis, adjusted for age and/or sex, was used to measure the relationship between the MEQ scores and physical functions (grip strength, MVPA, and appendicular skeletal mass), and MEQ scores and MMSE scores. Multivariate logistic regression analysis was used to calculate the odds ratio and 95% confidential interval of developing dementia in relation to chronotype and volume of physical activity. The MMSE score was labeled as a dependent variable, whereas chronotype and physical activity volume were labeled as the independent variables. Model 1 was the robust group; Model 2 was adjusted for age, sex, and BMI; and Model 3 was adjusted for age, sex, BMI, EQ-5D, and GDS in the logistic regression model.

## 3. Results

The mean age of participants was 77.0 years (±3.7 years). The demographic and clinical characteristics are interpreted in Table 1. The differences in weight, height, lean body mass, and grip strength between men and women were found to be significant, whereas BMI, physical activity, and mental health did not differ significantly.

The relationships between MEQ scores and physical functions, and MEQ scores and MMSE scores were analyzed using partial correlation analysis, and are explained in Table 2. Partial correlation analysis, controlled for age and sex, showed significant positive associations in the MMSE score (r = 0.27), MVPA (r = 0.32), and grip strength (r = 0.23). The participants aged more than 76 years showed stronger associations compared to participants aged 70 to 75 years.

Linear regression analysis was performed to examine the relationship between MEQ scores and physical activity in participants >75 years and ≤75 years of age. The participants with higher MEQ scores, morning types, showed a higher volume of physical activity in all participants. Moreover, this association was observed to be stronger in participants aged >75 years (r = 0.42, *p* < 0.005) than participants aged ≤75 years (r = 0.31, *p* < 0.05).

The differences in physical health and MMSE scores between the morning types (92) and evening types (78) are explained in Table 3. Significant statistical differences were observed in the preferred gait speed, grip strength, MVPA, and GDS. In the case of MMSE scores and some other physical functions like body mass index and lean body mass, no significant differences were found.

The chronotype and physical activity data were analyzed by multivariate logistic regression analysis to predict the estimated risk of dementia (Table 4). In Model 1, participants with the evening chronotype and low daily physical activity had 2.31 times and 2.51 times higher estimated risk of dementia respectively. The participants with both the evening chronotype and lower physical activity had a 3.67 times higher estimated risk of dementia. After adjustment for age, sex, and BMI in Model 2, and further adjusting for EQ-5D and GDS in Model 3, the odds were still present but lower than Model 1.

## 4. Discussion

In this study, we examined the relationships between the MMSE score, chronotype, and daily physical activity in nondemented community-dwelling older adults. As a result, we found that the evening chronotype, low daily physical activity, and MMSE scores were positively associated with each other. These relationships were particularly stronger in participants aged ≥76 years. Upon further examination, the estimated risk of dementia was observed to be 3.05 to 3.67 times higher in participants with low daily physical activity and the evening chronotype, and 1.95 to 2.26 times higher in participants with low daily physical activity and the morning chronotype than those with high daily physical activity and the morning chronotype.

It has remained unclear whether the MMSE score is independently associated with chronotype and daily physical activity in nondemented older adults. To our knowledge, this is the first study to demonstrate that morningness–eveningness and daily physical activity levels may be independently associated with MMSE scores in nondemented older adults. We found that the morning types had a higher MMSE score in comparison to evening types, but the difference was not statistically significant. The association between the MEQ score and MMSE score were stronger in older adults aged 75 years and above. Previous studies have also shown significantly higher MMSE scores in morning types [20] than in evening types. Although the physiological mechanisms of the association between diurnal preference and the MMSE score is unclear, animal studies have shown the underlying mechanisms in the association between disrupted circadian rhythms and risk of AD. For example, it has been demonstrated that loss of central circadian rhythms accelerates amyloid plaque accumulation, while loss of peripheral Bmal1 in the brain parenchyma increases the expression of ApoE and promotes fibrillar plaque deposition [28].

However, the findings on the association between morning preference and a higher MMSE score in this study are not consistent with the results of research by Fang et al. [20] where, without controlling for education, morning types were found to have significantly lower MMSE scores than evening types. They estimated the preference using actual habitual behavior patterns such as self-reported habitual mid-sleep timing on free days. Another previous study reported that phase advances in both bedtime and wake-up time were associated with a lower MMSE score [29]. Considering that disrupted sleep may be a factor that drives the progression of AD [30], the phase advance in sleep timing might be associated with disrupted sleep, rather than the phase of the circadian clock. Future studies of the associations between diurnal preference/phase of the circadian rhythm, objectively measured habitual sleep timing, and MMSE score are needed.

The morning chronotype has been reported to be more physically active in adolescents [31,32] and the evening chronotype has been associated with higher odds of none to low level of physical activity and higher sitting time in adults [33]. Our study results further broaden these findings as a significantly higher level of physical activity (≥3 METs) was observed in morning chronotype compared to evening chronotype in older adults aged ≥70 years. Previous studies have shown that higher levels of physical activity reduce cognitive decline [34] as well as the risk of dementia [21], and may also be an important factor to prevent them [35]. Physical activity is one of the modifiable lifestyle factors associated with the risk of dementia and cognitive impairment [36]. A meta-analysis of a prospective cohort study reported that physical activity reduces the risk of dementia by 28% [37]. In older adults, especially women, aged 85 years and above, being physically active is shown to reduce cognitive impairment by 88% [38].

Studies have suggested that individuals engaging in physical activity, especially leisure time activities, are able to endure AD pathology and have efficient cognitive networks providing cognitive reserve, which delays the onset of dementia [39]. A greater amount of physical activity has also been associated with increased grey matter volume, reducing the risk of dementia twofold [40]. Several neuroimaging studies have also reported that a higher amount of physical activity prevents or delays the onset of dementia by modifying the size of brain areas that atrophy and shrink in later life [41,42].

There are some limitations to be considered in our study. Firstly, this cross-sectional study could not establish cause-and-effect relationships between cognitive function, diurnal preference, and daily physical activity. Longitudinal studies or interventional studies may be required to test whether changes in diurnal preference and daily physical activity have any effect on cognitive function in older adults. The effect of chronotype on the timing of physical and cognitive function tests should also be considered in future studies. Second, the effect of food intake was not controlled for. Evening diurnal preference is associated with higher alcohol consumption [43,44] which may be a risk factor of cognitive decline [45]. In future studies regarding chronotype and dementia, the influence of diet such as alcohol intake may also be considered. Third, the current participants were all Korean older adults in a particular city. Studies in other populations may be required to clarify to what extent the present results can be generalized.

## 5. Conclusions

In this study, we observed an independent association of chronotype and physical activity with risk of developing dementia. The estimated OR of dementia risk was seen to be significantly higher when both evening preference and low physical activity were present than when they were present individually. Evening types also had reduced daily physical activity and lower cognitive function than morning types. Although in our study the physiological mechanism of the association between chronotype and estimated risk of developing dementia is unknown, results show that chronotype preference may be an indication of risk for dementia. Our study results also indicate that moderate-intensity physical activity in daily life is protective against the estimated risk of dementia. Future study should focus on sleep quality and diet along with chronotype and physical activity to devise protective strategies and interventions against the risk of dementia and cognitive decline.

## Figures and Tables

**Figure 1 ijerph-17-03701-f001:**
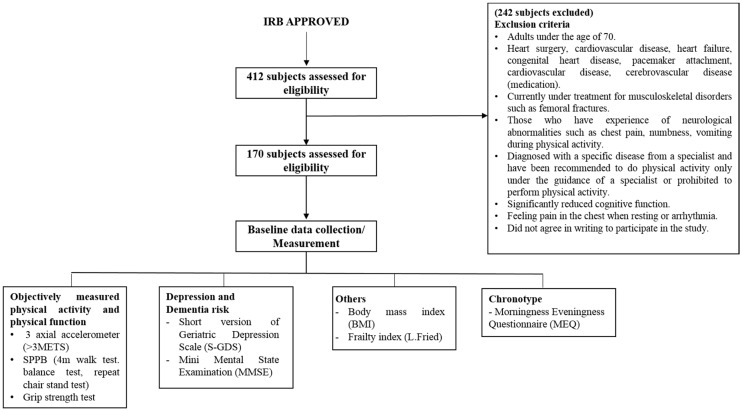
Flowchart explaining the screening procedure of participants, baseline data collection, and measurement methods.

**Table 1 ijerph-17-03701-t001:** Anthropometric, physical, and chronotype characteristics of the subjects.

Variables	Total	Women	Men
Number (n)	170	102	68
Age (years)	76.97 ± 3.69	76.63 ± 3.68	77.30 ± 3.70
Weight (kg)	62.12 ± 8.45	58.59 ± 8.01	65.64 ± 8.88 *
Height (m)	1.59 ± 0.05	1.53 ± 0.05	1.64 ± 0.05 *
Body mass index (kg/m^2^)	24.70 ± 3.28	24.87 ± 3.04	24.53 ± 3.53
Lean body mass (kg)	40.26 ± 4.47	35.18 ± 3.58	45.34 ± 5.36 *
Preferred gait speed (m/s)	1.15 ± 0.19	1.12 ± 0.18	1.19 ± 0.21
Grip strength (kg)	24.97 ± 4.53	20.19 ± 4.13	29.76 ± 4.93 *
Moderate-intensity physical activity (min)	11.75 ± 4.05	11.40 ± 3.90	12.10 ± 4.20
Morningness–Eveningness scores	56.44 ± 8.56	58.66 ± 7.93	54.21 ± 10.21
Mini-Mental State Examination	24.95 ± 3.51	24.63 ± 3.24	25.28 ± 3.78
Geriatric Depression Scale (15 items)	8.96 ± 3.82	8.90 ± 3.80	9.01 ± 3.83
Instrumental activities of daily living scale	10.94 ± 1.99	10.39 ± 1.35	11.49 ± 2.63
Falls efficacy scale	9.15 ± 1.40	8.90 ± 1.70	9.40 ± 1.10
Frailty scale score	2.07 ± 0.89	2.01 ± 0.91	2.12 ± 0.86

Variable values are given as mean ± SD; * versus men (*p* < 0.05) by Student’s *t*-test.

**Table 2 ijerph-17-03701-t002:** Partial (parametric and non-parametric) correlation coefficients (r) between Morningness–Eveningness scores, daily physical activity, Mini-Mental State Examination (MMSE) scores, and physical health.

	All	70–75 years	≥76 years
Morningness–Eveningness scores			
MMSE scores	0.27 *	0.23 *	0.32 *
MVPA	0.32 *	0.32 *	0.41 *
Grip strength	0.23 *	0.13	0.27 *
Appendicular skeletal muscle mass	0.19	0.16	0.20

* *p* < 0.05; Correlation coefficients were adjusted as appropriate for age and/or sex.

**Table 3 ijerph-17-03701-t003:** Physical and mental function by morningness–eveningness chronotype.

Variables	Morning Type	Evening Type
Number (male)	56 (36)	46 (32)
Age (years)	76.87 ± 3.15	77.03 ± 3.93
Body mass index (kg/m^2^)	23.19 ± 2.42	26.19 ± 2.93
Lean body mass (kg)	42.48 ± 2.98	38.04 ± 3.65
Preferred gait speed (m/s)	1.20 ± 0.19	1.08 ± 0.18 *
Grip strength (kg)	27.23 ± 3.13	22.68 ± 3.94 *
Moderate-intensity physical activity (min)	15.25 ± 2.86	8.22 ± 3.25 *
Morningness–Eveningness scores	74.12 ± 7.73	39.16 ± 8.58 ^†^
Mini-Mental State Examination	25.92 ± 3.15	23.99 ± 3.08
Geriatric Depression Scale 15 items	7.24 ± 2.51	10.68 ± 2.15 *

Variables are mean ± SD; * versus men (*p* < 0.05) by Student’s *t*-test; ^†^ versus men (*p* < 0.05) by Mann–Whitney U test.

**Table 4 ijerph-17-03701-t004:** Adjusted odds ratios (95% confidence intervals) for estimated dementia risk in the categories of chronotype and volume of daily physical activity (PA) in older adults.

	Model 1 Robust	Model 2	Model 3
MEQ chronotype (number)			
Evening types (78)	2.31 (1.30–2.91)	2.27 (1.25–2.67)	1.89 (1.00–2.86)
Morning types (92)	1	1	1
MVPA physical activity			
Low-dose, inactive (99)	2.51 (1.41–3.01)	2.33 (1.28–2.87)	1.93 (1.14–2.56)
High-dose, active (71)	1	1	1
MEQ chronotype and PA			
Low PA and Evening type (48)	3.67 (1.85–4.97)	3.51 (1.24–4.43)	3.05 (1.17–4.38)
Low PA and Morning type (51)	2.26 (1.12–3.03)	2.22 (1.00–2.94)	1.95 (0.99–2.97)
High PA and Evening type (30)	1.97 (0.98–2.15)	1.89 (1.00–2.02)	1.90 (0.89–1.97)
High PA and Morning type (41)	1	1	1
*p* trend	<0.05	<0.05	n.s.

Odds ratios (95% confidence intervals) adjusted for age, sex, health-related QOL, total sleep time, current smoking, and alcoholism, n.s.= not significant.

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
