# Peer review of "The Relationship between Chronotype, Physical Activity and the Estimated Risk of Dementia in Community-Dwelling Older Adults"

_ijerph, 2020, doi:10.3390/ijerph17103701_

Round 1

Reviewer 1 Report

The manuscript "The relationship between chronotype, physical activity and mental health in community-dwelling older adults" was aimed to characterize the association between chronotype, physical activity, and risk of dementia in older adults.

In my opinion the manuscript is well-written, interesting and useful to address further researches on the positive effects of sleep and physical activity on cognitive abilities.

The use of the accelerometer allowed an objective measure of physical activity.

The limitations of the study were adequately recognized.

Minor revision. Line 75: "under the" is repeated twice.

Author Response

Comment 1: Line 75: "under the" is repeated twice.

Response 1: Thank-you for reviewing our paper. We deeply appreciate your comments.

The correction has been made. We have proof-read the manuscript in depth.

Reviewer 2 Report

The study used methods and techniques to assess or measure the physical and mental state of older adults, but the key points and the logic of the manuscript cannot be clearly followed. The paper only shows there were associations between sleep behavior and mental state of older adults, but as the causal relationships of the variables were uncertain, so the conclusions cannot be used to predict the risk for developing dementia. The authors emphasized several times throughout the manuscript that chronotype and daily physical activity were predictors for the risk of having dementia, which are lack of basis and unconvincing. Title: It is“mental health”in the title, but the topic in the introduction section is all about “Alzheimer's disease.” What “community-dwelling”refers to? It is not explained in the introduction part. Abstract: 1. P1 L13, the sample size is “239”here, which is inconsistent with the number “170” in Figure 1 and other tables (Table1, Table 3). Please check. 2. There are many grammar mistakes in the abstract, especially for singular and plural. For example, the sentence “…our findings suggests that”should be “…suggest…” Introduction: 1. L24. The definition of “dementia”is given, but what is the relationship between dementia and mental health tested in the study? 2. L30. “AD”is abbreviated, please write the full name for the first time. 3. L25. what does“very five years”mean? 4. L27. “too”is redundant. 5. L38-43. the questionnaires mentioned should be cited, to let readers know their sources. 6. L56-64. Research questions are not clearly presented in this part. Materials and Methods: 1. L66-68. Again, what is the sample size? There are so many numbers, and the numbers in the text are not consistent with those in Figure 1. Please check. 2. L68-77. The sentences are the same as they are in Figure 1. So why write the text twice in the manuscript? 3. L80-89. Please give references for the questionnaires you used. 4. L79-119. The data collection information is stuffed in this section without organization. A Table can be provided to list the indexes, and show how they were measured or assessed. 5. L129-130. What is the dependent variable? Are they all dependent variables? 6. L131. The sentence“All participants signed informed written consent and the study procedure was approved by IRB.”should be mentioned earlier. 7. L133-135. Please check the numbers. 8. P120-130. Please explain why multiple logistic regression was used? Results: 1. The results with different themes are not well presented. This needs to be reorganized. 2. L143-150. What are the variables for partial correlation analysis? They are not clearly presented in text and in table. 3. L162. Table 3. Is“Number”also one of the variables? 4. L162-163. Table 3. The p should be lowercase and italicized. 5. L163-172. The results are unconvincing. You cannot predict the risk of dementia with the variables. Discussion: The section seems to lack focus around key issues.

Author Response

The study used methods and techniques to assess or measure the physical and mental state of older adults, but the key points and the logic of the manuscript cannot be clearly followed. The paper only shows there were associations between sleep behavior and mental state of older adults, but as the causal relationships of the variables were uncertain, so the conclusions cannot be used to predict the risk for developing dementia. The authors emphasized several times throughout the manuscript that chronotype and daily physical activity were predictors for the risk of having dementia, which are lack of basis and unconvincing.

Title

Comment 1:  It is “mental health” in the title, but the topic in the introduction section is all about “Alzheimer's disease.” What “community-dwelling” refers to? It is not explained in the introduction part.

Response 1: Thank-you for reviewing our paper. We are immensly greatful your comments. We have changed the world “ mental-health” with “risk of dementia” to reflect the contenets of the manuscript better. The title now reads as follows:

“The relationship between chronotype, physical activity and the estimated risk of dementia in community-dwelling older adults”.

The word “community-dwelling” is used in our study as all participants were from same community in Busan, South Korea. Several studies [1, 2] have used this term to describe the participants.

  1. Grossman, David C., Susan J. Curry, Douglas K. Owens, Michael J. Barry, Aaron B. Caughey, Karina W. Davidson, Chyke A. Doubeni et al. "Interventions to prevent falls in community-dwelling older adults: US Preventive Services Task Force recommendation statement." Jama 319, no. 16 (2018): 1696-1704.
  2. Liu-Ambrose, Teresa, Jennifer C. Davis, John R. Best, Larry Dian, Kenneth Madden, Wendy Cook, Chun Liang Hsu, and Karim M. Khan. "Effect of a home-based exercise program on subsequent falls among community-dwelling high-risk older adults after a fall: a randomized clinical trial." Jama 321, no. 21 (2019): 2092-2100.

Abstract

Comment 1: P1 L13, the sample size is “239”here, which is inconsistent with the number “170” in Figure 1 and other tables (Table1, Table 3). Please check.

Response 1: Thank-you for the comment. The correction has been made.

Comment 2: There are many grammar mistakes in the abstract, especially for singular and plural. For example, the sentence “…our findings suggests that” should be “…suggest…”

Response 2: Thank-you for reviewing our paper. We deeply appreciate your comments.

The correction has been made. We have proof read the manuscript in depth.

Introduction

Comment 1: L24. The definition of “dementia” is given, but what is the relationship between dementia and mental health tested in the study?

Response 1:  Thank-you for the comment.This comment was extremely hepful to make the idea of our manucript more better.

We have modified title to reflect our study more clear i.e. to investigate the independent associations of the diurnal preference (Chronotype) and daily physical activity with the estimated risk of dementia in nondemented community-dwelling older adults . The title now reads as,  “The relationship between chronotype, physical activity and the risk of dementia in community-dwelling older adults”.

Comment 2: L30. “AD” is abbreviated, please write the full name for the first time.

Response 2: Thank-you for the comment. The correction has been made.

Comment 3: L25. What does “very five years” mean?

Response 3: Thank-you for the comment. It was a spelling mistake. The correction has been made.

Comment 4: L27. “too” is redundant.

Response 4: Thank-you for the comment. The correction has been made.

Comment 5: L38-43. the questionnaires mentioned should be cited, to let readers know their sources.

Response 5: Thank-you for the comment. The citation has been added.

Comment 6: L56-64. Research questions are not clearly presented in this part.

Response 6: Thank-you for the comment. We have extendend and modified the study objective as follows:

“Thus, in this study, we aimed to investigate the independent associations of the diurnal preference (Chronotype) and daily physical activity with the estimated risk of dementia in nondemented community-dwelling older adults. We also investigated the combined effect of chronotype and physical activity with the estimated risk off developing dementia.”

Materials and Methods

Comment 1: L66-68. Again, what is the sample size? There are so many numbers, and the numbers in the text are not consistent with those in Figure 1. Please check.

Response 1: Thank-you for the thorough review of our paper and the comment.

The subject number has been corrected in the methods section.

Comment 2: L68-77. The sentences are the same as they are in Figure 1. So why write the text twice in the manuscript?

Response 2: Thank-you for the comment. We agree with the reviewer and have removed the exclusion criteria from the method section.

Comment 3: L80-89. Please give references for the questionnaires you used.

Response 3: Thank-you for the comment. We have added references to all the questionnaires used in our study in the manuscript.

Comment 4: L79-119. The data collection information is stuffed in this section without organization. A Table can be provided to list the indexes, and show how they were measured or assessed.

Response 4: Thank-you for the comment. We have organized the method section with sub-headings to provide clear information of the data collection.

Comment 5: L129-130. What is the dependent variable? Are they all dependent variables?

Response 5: Thank-you for the comment.

The dependent and independent variables are described in the methods section as:

The risk of dementia was labeled as a dependent variable whereas chronotype and physical activity volume was labeled as the independent variable.”

Comment 6: L131. The sentence “All participants signed informed written consent and the study procedure was approved by IRB.” should be mentioned earlier.

Response 6: Thank-you for the comment. We have moved the sentence to line 70-71 and now reads as:

“All participants signed informed written consent and the study procedure was approved by Institutional Review Board (IRB).”

Comment 7: L133-135. Please check the numbers.

Response 7: Thank-you for the comment. The correction has been made.

Comment 8: P120-130. Please explain why multiple logistic regression was used?

Response 8: Thank-you for the comment. We have modified and added the use of multiple logistic regression in the manuscript as follows:

“Multiple logistic regression models were used to assess the estimated risk of dementia in relation to chronotype and volume of physical activity. The MMSE score was labeled as a dependent variable whereas chronotype and physical activity volume was labeled as the independent variable. Model 1 was the robust group, Model 2 was adjusted for age, sex, and BMI, and Model 3 was adjusted for age, sex, and BMI, EQ-5D and GDS in the logistic regression model.”

Results

Comment 1: The results with different themes are not well presented. This needs to be reorganized.

Response 1: Thank-you for the comment. The correction has been made.

Comment 2: L143-150. What are the variables for partial correlation analysis? They are not clearly presented in text and in table.

Response 2: Thank-you for the comment. We have modified the sentence to make the variables used for partial correlation analysis more clear and now it reads as follows:

“Partial correlation analysis, adjusted for age and/or sex, was used to measure relationship between MEQ scores and physical functions (Grip strength, MVPA and appendicular skeletal mass), and MEQ scores and MMSE scores.”

Comment 3: L162. Table 3. Is “Number” also one of the variables?

Response 3: Thank-you for the comment. The number is not the variable. We have removed it from the table to avoid confusion.

Comment 4: L162-163. Table 3. The p should be lowercase and italicized.

Response 4: Thank-you for the comment. The correction has been made.

Comment 5: L163-172. The results are unconvincing. You cannot predict the risk of dementia with the variables.

Response 5: Thank-you for the comment. We have modified ther results as follows:

“The chronotype, physical activity data were analyzed by multivariable models using Logistic regression analysis to predict the odds of developing dementia (Table 4). In model 1, participants with evening chronotype and low daily physical activity had 2.31 times and 2.51 times higher odds of developing dementia, respectively. The participants with both evening chronotype and lower physical activity had a 3.67 times higher odds of acquiring dementia. After adjustment for age, sex, and BMI in model 2, and further adjusting for EQ-5D and GDS in model 3, the odds were still present but lower than model 1.”

Discussion

Comment 1: The section seems to lack focus around key issues.

Response 1: Thankyou for the comment. We have modified and extended the discussion section as follows:

“The morning chronotypes have been reported to be more physically active in adolescents [31, 32] and evening chronotypes have been associated with higher odds of none to low level of physical activity and higher siting time in adults [33]. Our study results further broadens this data as significantly higher level of physical activity (≥3 METS), was observed in morning chronotypes compared to evening chronotypes in older adults aged ≥70 years. Previous studies have shown that higher levels of physical activity reduce cognitive decline [34] as well as the risk of dementia [21] and also may be an important factor to prevent cognitive decline and dementia [35]. Physical activity is one of the modifiable lifestyle factors associated with risk the of dementia and cognitive impairment [36]. A meta-analysis study of a prospective cohort study reported that physical activity reduces the risk of dementia by 28% [37]. In older adults, especially women, aged 85 years and above being physically active is shown to reduce cognitive impairment by 88% [38].

Studies have suggested that individuals engaging in physical activity, especially leisure time activities, are able to endure AD pathology and have efficient cognitive network providing cognitive reserve which delays onset of dementia [39]. A greater amount of physical activity has also been associated with increased grey matter volume reducing the risk of dementia by 2 folds [40]. Several neuroimaging studies have also reported higher amount of physical activity prevents or delays onset of dementia by modifying size of brain areas that atrophy and shrink in later life [41, 42].”

Reviewer 3 Report

The article is very well prepared. The group of people examined is representative.
Very detailed experimental studies were conducted. I have no comments on the methodology.
A well conducted discussion. There are several other factors, not mentioned in the article, that may affect MMSE, but the authors are aware of them. They indicated them in the limitations of the work, at the end of the discussion. Literature items were properly selected, most of them published after 2000.

However, I would like to ask you some questions, which can be referred to in the article:
1. what is the gender participation in the morning and evening chronotype?
2. what time of day were fitness/balance/walking tests etc. performed? Perhaps the timing of these tests in older people with different chronotypes will influence the result?

Author Response

Comment 1. What is the gender participation in the morning and evening chronotype?

Response 1:  Thaankyou for reviewing our mauscript. We are immensly garateful for your construcitve comments which help us to improve our paper.

 In  morning chronotype the no. of females were 56 and no. of males were 36. In evening chronope the  no. of females and males were 46 and 32 respectively. We have added this information in the paper in table 3. We did not observe significant gender differences in our study which is consistent with results reported by previous studies [1,2].

References

  1. Roenneberg, Till, Tim Kuehnle, Myriam Juda, Thomas Kantermann, Karla Allebrandt, Marijke Gordijn, and Martha Merrow. "Epidemiology of the human circadian clock." Sleep Medicine Reviews 11, no. 6 (2007): 429-438.
  2. Randler, Christoph, and Judith Engelke. "Gender differences in chronotype diminish with age: A meta-analysis based on morningness/chronotype questionnaires." Chronobiology international 36, no. 7 (2019): 888-905.

Comment 2. What time of day were fitness/balance/walking tests etc. performed? Perhaps the timing of these tests in older people with different chronotype will influence the result?

Response 2: Thankyou for throoughly reviewing our paper. We are grateful for your comments. It helped us immensly to improve our paper.

The physical function tests were carried out during the afternoon time (13:00 pm to 17:00 pm). However the effect of chronotype on physical fucntion test result was not measured. It is also one of the limitation of our study and we have mentioned it in the manuscript as, “The effect of chronotype on the timing of physical and cognitive function tests should also be considered in future studies.”

Round 2

Reviewer 2 Report

The manuscript has been improved, but still there are some grammatical and typographical errors, please check it carefully.

For example:

L16, “The Korean version… were…”should be “The Korean version… was…”

L86, “Depressive symptom was also assed…”what does assed mean?

L91-92, Please rephrase this sentence to make it clear: “Physical activity (PA) was objectively measured through monthly-averaged daily moderate to 91 vigorous PA (MVPA) of intensity over 3 METs was measured with a tri-axial accelerometer.”

L181, Table 4, How many decimal numbers are there? In the first line of Model 2, it is “2.27(1.251-2.67).”

L168, It should be “Table 3” instead of “table 3.”

Some other comments:

L124-125, please state clearly why Student’s t-test was used, and for what? To analyze the difference between males and females?

L174-175, what is “multivariable models using logistic regression analysis?” Multinomial logistic regression? Binary logistic regression? Or, ordered logistic regression? What the data format of the dependent variable (risk of dementia)? Please provide the information.

L183, Table 4, notes: “n=E 78, M 92”It is not very understandable, please make it clear.

Author Response

Comment 1: The manuscript has been improved, but still there are some grammatical and typographical errors, please check it carefully.

Response 1: We thank the reviewer for the detailed and thorough comments.

we have carefully checked our manuscript, and we also   have used “Grammarly” software to check grammatical, typographical, and spelling errors.

For example:

Comment 2: L16, “The Korean version… were…”should be “The Korean version… was…”

Response 2: The correction has been made as your comment.

(line 16: were to was)

Comment 3: L86, “Depressive symptom was also assed…”what does assed mean?

Response 3: The typographical error has been corrected.

(line 86; “Depressive symptom was also assed” to “Depressive symptom was also assessed”)

Comment 4: L91-92, Please rephrase this sentence to make it clear: “Physical activity (PA) was objectively measured through monthly-averaged daily moderate to 91 vigorous PA (MVPA) of intensity over 3 METs was measured with a tri-axial accelerometer.”

Response 4: Thank you for the comment. The sentence has been rephrased and now reads as (line 92):

“Daily physical activity (PA) was objectively measured through moderate to vigorous PA (MVPA) of intensity over 3 METs with a tri-axial accelerometer (Fitmit INC, Suwon, South Korea).”

Comment 5: L181, Table 4, How many decimal numbers are there? In the first line of Model 2, it is “2.27(1.251-2.67).”

Response 5: The correction has been made.  (“1.251” to “1.25”)

Comment 6: L168, It should be “Table 3” instead of “table 3.”

Response 6: The correction has been made.  (“table 3” to “Table 3”)

Comment 7: L124-125, please state clearly why Student’s t-test was used, and for what? To analyze the difference between males and females?

Response 7: Thank you for the comment. We have expanded the information and the sentence now reads as (line 124):

“Student’s t-test analyzed the difference in baseline variables (demographic, anthropometric, physical functions, MMSE scores, and chronotype) between men and women.”

Comment 8: L174-175, what is “multivariable models using logistic regression analysis?” Multinomial logistic regression? Binary logistic regression? Or, ordered logistic regression? What the data format of the dependent variable (risk of dementia)? Please provide the information.

Response 8: Thank you for the comment.

In our study we used bivariate dependent variable (have the risk of dementia and have not the risk of dementia), and two and four independent variable (2 chronotype variable “evening & morning” and/or 2 volume of physical activity variable “Low & High”) to predict the odds of developing dementia, hence we used multivariate logistic regression analysis.

In the Elsevier Health science, the Multivariate logistic regression analysis is defined as an extension of bivariate regression in which “two or more independent variables are taken into consideration simultaneously” to predict a value of a dependent variable for each subject [1].

Therefore,  we change the in the statistical analysis (line 133)“Multivariate logistic regression analysis were used to calculate the odds ratio and 95% confidential interval of developing dementia in relation to chronotype and volume of physical activity.”

And the results section (line 177) “ The chronotype, physical activity data were analyzed by multivariate logistic regression analysis to predict”

Reference

  1. Liu, Longjian. Heart Failure: Epidemiology and research methods. Elsevier Health Sciences, 2017.

Comment 9: L183, Table 4, notes: “n=E 78, M 92”It is not very understandable, please make it clear.

Response 9: we added the subject number in each group in the Table 4.
